# The Structures, Functions, and Roles of Class III HDACs (Sirtuins) in Neuropsychiatric Diseases

**DOI:** 10.3390/cells13191644

**Published:** 2024-10-02

**Authors:** Robin E. Bonomi, William Riordan, Juri G. Gelovani

**Affiliations:** 1Department of Psychiatry, Yale University, New Haven, CT 06511, USA; william.riordan@yale.edu; 2College of Medicine and Health Sciences, Office of the Provost, United Arab Emirates University, Al Ain P.O. Box 15551, United Arab Emirates; jgelovani@uaeu.ac.ae; 3Department of Biomedical Engineering, College of Engineering and School of Medicine, Wayne State University, Detroit, MI 48201, USA; 4Department of Radiology, Division of Nuclear Medicine, Siriraj Hospital, Mahidol University, Bangkok 10700, Thailand

**Keywords:** HDACs, neuropsychiatric, sirtuins

## Abstract

Over the past two decades, epigenetic regulation has become a rapidly growing and influential field in biology and medicine. One key mechanism involves the acetylation and deacetylation of lysine residues on histone core proteins and other critical proteins that regulate gene expression and cellular signaling. Although histone deacetylases (HDACs) have received significant attention, the roles of individual HDAC isoforms in the pathogenesis of psychiatric diseases still require further research. This is particularly true with regard to the sirtuins, class III HDACs. Sirtuins have unique functional activity and significant roles in normal neurophysiology, as well as in the mechanisms of addiction, mood disorders, and other neuropsychiatric abnormalities. This review aims to elucidate the differences in catalytic structure and function of the seven sirtuins as they relate to psychiatry.

## 1. Introduction

Histone deacetylases (HDACs) are a family of 18 enzymes, classified into four groups (I–IV) with wide-ranging implications in health and disease. Class III HDACs are comprised of HDAC enzymes termed “silent information regulators” (sirtuins, SIRTs), representing a family of closely related deacetylases that are Zn^2+^-independent but NAD^+^-dependent. The SIR2 (silent information regulator) gene in the murine model [1] led to further understanding of seven sub-types of SIRTs; SIRTs 1–7 have been identified in humans [2]. Sirtuins are implicated in a variety of cellular processes, including gene silencing [1,3], cell cycle regulation [4], metabolism under energy deficits [5,6,7,8], apoptosis [9,10], and longevity [5,11,12,13], as well as mood, anxiety, and aging [14,15]. SIRT isoforms vary in their cellular localization, with SIRTs 1, 6, and 7 being primarily nuclear bound and SIRTs 3, 4, and 5 being mitochondrial. While sirtuins cleave the acetyl moiety from a lysine residue, it is understood that some sirtuin isoforms can cleave larger leaving groups than the acetyl moiety (i.e., glutaryl, succinyl, myristoyl, etc.) [16]. Within this larger role, the sirtuins often act on transcription factors and other cellular regulatory proteins rather than solely on histone core proteins, as the broader name might suggest [17,18].

Early investigations into SIRT1, the original SIR2 homolog, uncovered the critical role in longevity and calorie restriction [9,12]. Pharmacologic agents such as resveratrol took the health industry and research by storm, opening a window into extending the human lifespan [8,13,19,20,21,22,23]. This expanded into studies of mood, anxiety, diabetes, and related illnesses [24]. However, much like other epigenetic mechanisms, the early promise of SIRT1 activation did not lead to a cure-all as many original papers once indicated it might. Neurodegenerative disease researchers (i.e., Alzheimer’s) and others interested in mood and anxiety further studied SIRT1, 2, and 3, uncovering crucial roles for these in neuropsychiatric illnesses and the promise for targeted therapeutic interventions given their roles in metabolic and inflammatory pathways [25,26]. SIRT5, a mitochondrial enzyme with implications on cellular energy and downstream mood effects [6,27,28], has also been a furtive target of study in terms of its role in conserving energy stores within the brain (Figure 1). Some of the lesser-known sirtuins, 4, 6, and 7, may play pivotal roles in substance use disorders.

Targeting epigenetic enzymes with psychiatric medications has been well-documented in other instances. Valproic Acid (VPA) is one of the most used mood stabilizers in psychiatry and has additional uses in neurology as an anti-epileptic. VPA primarily targets histone deacetylases; however, it does not inhibit sirtuins, only HDAC classes I, II, and IV [29,30,31]. Though VPA has been trialed in fear extinction and post-traumatic stress disorder [32], it has not yet proven successful [33]. Similarly, sodium butyrate has garnered attention for targeting histone deacetylases in psychiatric treatment but does not inhibit sirtuins [34,35,36]. Treatment for PTSD has extensively sought out epigenetic-targeted treatments, but it stands to reason that sirtuins should be viewed differently as they are structurally and functionally quite different from other HDACs [37].

This review aims to tie together the existing literature on the structure and function of each sirtuin within psychiatric illness and the evidence for targeting sirtuins in disease. The significant structural similarity among sirtuins makes targeting individual enzymes with pharmacologic agents challenging. Much of the data for efficacy in the literature becomes confounded by cross-sirtuin inhibition or activation with one agent. Further research is imperative if we are to better understand each individual enzyme and its unique role in health and pathology.

## 2. Insights into Sirtuin Catalytic Mechanisms

All sirtuins contain a large Rossman fold (seven parallel beta sheets connected by an alpha helix) encompassing the NAD^+^ binding domain and a smaller, more variable zinc ion binding domain. While SIRT enzymes have retained the Zn^2+^ binding domain from the other classes of HDACs, this domain is not utilized as the primary coordination site for catalytic cleavage as it is for other HDACs. Within the SIRTs, there are mechanistic differences, including one group of mono-adenosine dinucleotide phosphate (ADP) ribosyl transferase, largely grouped as ‘ART’ functioning enzymes. The other encompasses the nicotinamide dinucleotide (NAD^+^)-dependent deacetylase family termed ‘DAC’ [1]. Wherein SIRTs 1, 2, and 3 are able to function as both ARTs and DACs to bind the substrate prior to the NAD^+^ [38]. SIRTs 4, 5, 6, and 7 act as NAD^+^-dependent allosteric activator enzymes, only DAC. This indicates that they rely on the binding of NAD^+^ prior to substrate binding and are unable to act as an ART alone.

To review the cleavage mechanism for sirtuins briefly, as it has been well described previously [39,40,41,42], the acetylated lysine chain enters the active site from a channel opposite to the NAD^+^, where the lysine-leaving group forms a transient attachment to adenine [43] (Figure 2). The oxygen positioned at the leaving group carbonyl attacks the 1′ position of the adenine sugar, which is activated by a histidine or tyrosine residue [44]. The 2′-hydroxyl group is subsequently deprotonated by the histidine or tyrosine residue [41]. As the nicotinamide leaves, the lysine–acetyl moiety is transiently connected to the 1′ position through an ester linkage. An oxocarbenium ring forms when the 2′ hydroxyl oxygen attacks the double nitrogen amide bond. This oxocarbenium intermediate is stabilized by the presence of a highly conserved asparagine residue in all sirtuins [44]. When the oxocarbenium ring opens, the lysine leaves with a free amino terminus while the acetyl (or other leaving group) moiety is transferred to the 2′ position of adenine dinucleotide and is connected via an ester linkage. Two histidine residues facilitate this leaving group transition from the 1′ position to the 2′ position on the adenine ribose ring [44]. The ester linkage of the leaving group to the ADPR has been observed in cases of the long acyl chain leaving group as well as the small acetyl moiety [18].

Within the four active site loops, there are large and small domains of the protein consisting of alpha helices and loops [45]. The large domain residues, which include the NAD+ binding site, are identical across Sirt1, 2, 3, 4, 5, and 6; however, the small domains, including alpha helices 5, 6, and 9, and loops 3 and 4 (L3 and L4), vary [45]. All sirtuins are made up of a Rossman fold containing the small loop where the active substrate binds, termed the “FGE” loop after the highly conserved FGExL that is contained within [46]. These allow for beta-sheet interactions with the substituted amino terminus of the target lysine protein and further differentiate the client proteins of each sirtuin. An excellent review of the wide-ranging structure and epigenetic targets of the seven sirtuins has previously been published by H. Jing and H. Liu [47].

## 3. SIRT1

### 3.1. Structure and Function

SIRT1 is an unordered protein due to a flexible loop involved in the active C-site residues, forming about half of the total sequence [48], and is predominantly known for its role in longevity and lifespan, though this has been recently expanded into anxiety and depression. Its sensitivity to cellular glucose levels [8] lend itself to implications in psychiatry beyond longevity and extending lifespan. While the core in all sirtuins is made from a Rossman fold, containing a channel terminating near the NAD^+^ ribose ring, the SIRT1 active site contains five aromatic residues, Tryptophan (Trp)176, Tyrosine (Tyr)185, Phenylalanine (Phe)187, Trp221 and Trp624, which contribute to pi-stacking interactions [48]. These aromatic residues line the hydrophobic pocket where the acetylated (or other moieties) lysine binds and could indicate that aromatic ring-leaving groups with hydrophobic features would bind well in the pocket and contribute to the pi–pi stacking [48]. Interestingly, SIRT1 is unable to effectively cleave propionyl or butyryl groups and functions at only ~28% and ~2%, respectively, of its deacetylase efficacy in these cases [49,50]. However, SIRT1 dephenylacetylation catalysis is about 56% of the deacetylation rate [51]. Other studies successfully developed and visualized SIRT1 expression activity through the utilization of a small molecule radioligand with fluorine substituted benzene leaving group on a lysine mimetic [52] and positron emission tomography imaging. Using lysine debenzoylation or dephenylacetylation allows for SIRT1 specificity, as most other sirtuins are unable to cleave this moiety [52,53]. The importance of the linker region is also highlighted. In SIRT1, the distance between the a-carbon and the side chain acetamido group plays a large role in the catalytic efficiency of SIRT1; thus, the full lysine chain is necessary for the activity [51]. Furthermore, the lysine center carbon conformation (L-conformation) is also extremely important, as all catalytic activity is lost from the enzyme with the D-lysine isomer [54].

SIRT1 catalytic activity is regulated by the phosphorylation of key functional site residues (i.e., threonine (T), serine (S), and tyrosine (Y) residues) on the enzyme surface. In SIRT1, the primary sites of regulation include S27, S47, T530, T540, T522, S434, and S682 (Protein Data Bank). Janus kinase 1 (JNK1) is the primary phosphorylation source for SIRT1, acting on S47 on hSIRT1 to increase SIRT1 activity upon increases in cellular glucose levels. Once the insulin levels rise in concert with elevated cellular glucose, SIRT1 is ubiquitinated and tagged for degradation [55]. Janus kinase 2 (JNK2) antagonizes JNK1 to stabilize SIRT1 through S27 phosphorylation and activates apoptotic pathways when a cell experiences damaged nuclear DNA [56]. Similarly, two members of the dual specificity tyrosine phosphorylation regulated kinase family, DRYK 1A and 3, phosphorylate SIRT1 at T522 to inhibit apoptotic pathways [57]. The ratio of cAMP/AMP also alters SIRT1 activity, particularly when responding to cellular glucose levels, as cAMP regulates protein kinase A (PKA) to phosphorylate S434 and enhance activity. This promotes fatty acid oxidation under conditions of glucagon receptor stimulation in the liver [58]. The same pathway is also stimulated by oleic acid to activate the SIRT1–PGC1α complex in skeletal muscles to increase rates of fatty acid oxidation [59]. SIRT1 is dependent on NAD^+^ for catalytic activity. Therefore, the cellular ratios of NAD^+^/NADH within a cell regulate SIRT1 catalytic function. Under energy deprivation conditions, the ratio of NAD^+^/NADH will increase, thus increasing SIRT1 activity. SIRT1 is endogenously activated with calorie restriction, which may contribute to an increase in lifespan and longevity [5]. Accordingly, genes associated with cellular death and apoptosis are also suppressed by SIRT1 via deacetylation of histone core proteins 1, 3, and 4 (H1, H3, and H4) at H1K26, H3K9, H4K16, and H4K56, [60] and the regulation of RNA polymerase I transcriptional machinery.

### 3.2. Psychiatric Implications

Primarily a SIRT1 activator, resveratrol has garnered attention across disciplines as a therapeutic agent. However, subsequent studies produced mixed results for efficacy. A polyphenol belonging to the family of stilbenes, resveratrol (3,5,4-trihydroxystylbene) is predominantly found in grapes and red wine and has become one of the most well-studied agents for increasing lifespan and longevity [61]. Studies previously demonstrated prolonged lifespan with resveratrol administration and specifically within calorie restriction, but these results have not been subsequently replicated or translated to clinical populations successfully [23]. Resveratrol as a sirtuin activator has been trialed for everything from increasing lifespan to reducing obesity to the treatment of mood and anxiety illnesses, though none have proven to be a clear area for treatment.

Preclinical evidence demonstrates a protective role for resveratrol in the hippocampus following sleep deprivation, lessening the cognitive deficits experienced [62].

Within psychiatry, global increases in SIRT1 activity may aid in reducing obesity [63] and thereby reduce depressive mood symptoms, improve longevity, and decrease all-cause mortality [64,65,66,67]. Interestingly, the converse is also true, where endogenous overactivation of SIRT1 may be a driver in the pathophysiology of anorexia-nervosa [68]. Within anorexia-nervosa, an individual’s restrictive eating and weight loss becomes pathologic to the extent that it is difficult to stop due to a desire for control and often co-morbid mood and anxiety. Therefore, inhibition of SIRT1 may provide a novel method of intervention for this otherwise difficult disease to treat.

#### 3.2.1. Depression

Systemic treatment with the SIRT1 inhibitor EX-527 also demonstrated efficacy at protection in rodent models of anxiety and depression [24,69]. Evidence such as the above highlights the difficulty in studying these enzymes and leaves future pharmacologic agents with unclear direction. To better understand the individual role of SIRT1 in regions of the brain, preclinical studies utilized regional activation and inhibition of SIRT1 via direct injection. Direct injection of resveratrol into the nucleus accumbens produces an anxiogenic effect rather than an anxiolytic effect [70,71]. Interestingly, SIRT1 activation within the hippocampus and dentate gyrus blocks the formation of these behaviors following chronic unpredictable stress and aberrant dendritic structures [24]. It follows then that direct inhibition of SIRT1 within the area of memory formation in the hippocampus produces anxiety. Across other regions of the brain, SIRT1 activation and overexpression hold promise for the improvement of depression and anhedonia. Other studies report positive effects on mood with resveratrol treatment and SIRT1 activation. For example, resveratrol was noted to improve depressive-like behaviors in mice after maternal separation post-partum [71].

Within the medial prefrontal cortex, SIRT1 impacts depression-related behaviors through glutamatergic neurons in a sex-specific manner. Selective knockout studies suggest SIRT1 activity in the forebrain excitatory neurons regulates anhedonic behaviors via altering synaptic transmission and excitability of pyramidal neurons [72]. SIRT1 deacetylates the brain-specific helix loop helix transcription factor (NHLH2) at lysine 49 to activate the monoamine oxidase-A (MAO-A) promoter [73]. Liebert et al. (2011) [73] reported that brain-specific SIRT1 knockout mice exhibited decreased anxiety-like behaviors, which could be reversed with MAO inhibitors. These findings indicate a positive correlation between anxiety levels and SIRT1 activity. However, this has been opposed in data from other specific brain regions where increases in SIRT1 lead to decreasing anxiety. Within the basal nucleus of the stria terminalis, for example, SIRT1 overexpression decreased anxiety behaviors through normalization of the corticotropin-releasing hormone [74]. Neuropeptide Y, another critical regulator of hormones, longevity, and appetite, has been linked to SIRT1 within the brain of the depressed rodent model line [75]. These findings suggest that SIRT1 plays a crucial role in modulating anxiety, though the directionality across different brain regions is unclear.

A study by Zocchi and Sassone-Corsi [76] showed that SIRT1 modulates the circadian rhythm. It is established that circadian rhythms and sleep play a role in mood and anxiety stability; this study highlights a role for SIRT1 in linking these two. Pharmacological activation of SIRT1 with resveratrol produces antidepressant-like effects in rodent models. Furthermore, Gao et al. [77] demonstrated that SIRT1 activation alleviated depressive-like behaviors induced by chronic stress in mice by reducing neuroinflammation and enhancing neurogenesis in the hippocampus. The use of *S*-Ketamine, an FDA-approved medication for treatment-resistant depression, in rodents following chronic unpredictable stress produces increased brain-derived neurotrophic factor (BDNF) and SIRT1 [78]. Further evidence points to a positive correlation between BDNF levels and SIRT1 activation, where the co-occurrence promotes cognitive function and lessens the impact of detrimental effects such as toxins, sleep debt, or models of depression and schizophrenia [62,70,71,79,80]. Studies demonstrate ketamine’s anti-depressant effects may be improved and lengthened through the use of a potent mTOR inhibitor, rapamycin [81,82]. SIRT1 inhibits mTOR inhibitor via TSC2 and direct inhibition of the mTOR complex formation [83,84]. Therefore, it could be hypothesized that SIRT1 synergistically enhances ketamine function like rapamycin. SIRT1 activators could, therefore, be an alternative target for enhancing and prolonging the acute effects of ketamine.

#### 3.2.2. Psychosis and Mania

Bipolar disorder is characterized by mood swings ranging from manic to depressive episodes. SIRT1 has been suggested to play a role in mood regulation and energy homeostasis, which are crucial in bipolar disorder. Pharmacological activation of SIRT1 improved mood stabilization and cognitive function in animal models [85,86]. Additionally, SIRT1 activation has been shown to influence mitochondrial function and oxidative stress, which are implicated in the pathophysiology of bipolar disorder [85]. Similarly, resveratrol and subsequent SIRT1 activation have demonstrated positive effects in protecting against cognitive impairments in a young mouse schizophrenia model [79].

Negative or anhedonia symptoms in schizophrenia are notoriously difficult to treat. Through SIRT1 interactions with p53 and p73, SIRT1 indirectly influences neuronal cell migration, nerve fiber outgrowth, growth cone motility, and axonal regeneration [87,88].

#### 3.2.3. PTSD

Metabolomic studies of individuals with post-traumatic stress disorder [32] show increased plasma levels of pyruvate, lactate, and glucose, with reduced plasma levels of citrate. Additionally, increased plasma levels of acylcarnitine suggest a reduced ability for mitochondrial fatty acid oxidation [89]. These findings highlight a dysfunction within the mitochondria of those with PTSD. Resveratrol, as well as other SIRT1 targeted agents, including quercetin and 5-aminoimidazole-4-carboxamide riboside (AICAR), are also in the process of being tested for novel therapeutics in PTSD [90]. Similarly, SRT2104, a selective SIRT1 activator, has been assessed in clinical trials for use as an anti-inflammatory agent to target depression and PTSD. In a randomized, double-blind, placebo-controlled study of healthy subjects, SRT2104 was effective at lowering levels of acute inflammation after lipopolysaccharide injection, a well-known proinflammatory agonist [91]. This was further confirmed through reductions in pro-inflammatory cytokines, including IL-6, IL-8, and C-reactive protein in the group receiving SRT2104. These findings hold promise that SIRT1 targeted agents could be helpful in treating the pro-inflammatory systemic effects of major depressive disorder (MDD) [15].

#### 3.2.4. Substance Use Disorders

Many substances, which are often misused and classified as psychiatric substance use disorders in the Diagnostic and Statistics Manual-5 (DSM-5), including amphetamines, opioids, cocaine, and alcohol, directly alter sirtuin levels in the brain. [92,93] demonstrated that cocaine increases the nuclear localization of SIRT1 in astrocytes selectively but not microglia. However, this localization was normalized through the use of piracetam, a cholinergic and glutamatergic agonist [93]. Several studies have found cell-wide increases in SIRT1 expression in the nucleus accumbens of mice chronically exposed to cocaine, heroin, or morphine compared to controls [93,94,95]. Interestingly, microinjection of a potent SIRT1 inhibitor, EX-527, into the ventrolateral orbital cortex, the region of the brain involved directly in drug reward-seeking behavior, reduced morphine use in rodents [96]. Viral (Herpes simplex virus) overexpression of SIRT1 increases indirect behavioral measures of drug reward to many substances in a dose-dependent fashion. This mimics cocaine’s upregulation of nucleus accumbens dendritic spine density, while the knockdown of SIRT1 decreased behavioral reward responses [94]. Taken together, it can be inferred that elevated SIRT1 activity and expression, through compensatory drug-induced upregulation and increased neuronal signaling in the mesolimbic and mesocortical pathways, may contribute to opioid addiction. Many of these studies found similar results for SIRT2, as well [93].

An alternative role has been elucidated for SIRT1 in alcohol use disorder where increased SIRT1 expression, via dihydromyricetin and resveratrol, were protective against deleterious effects of alcohol in the liver and the brain [97,98]. Additionally, alcohol-induced degeneration, including effects on mitochondrial lipid transport, inflammatory responses, and oxidative stress, was also ameliorated with SIRT1 overexpression [97,99]. A SIRT1 homolog, SIR2, in the fruit fly *Drosophila*, is critical to ethanol sensitivity and may be required for acquiring tolerance as well as alcohol preference [98]. Though additional research is needed to further elucidate the relationship between conditioned substance use and the role of SIRT1, especially in the brain, SIRT1 shows promise as an interventional target to mitigate the toxic effects of substance use on the body and potentially treat the symptoms of substance use disorders.

## 4. SIRT2

### 4.1. Structure and Function

Multiple crystal structures identified for SIRT2 [45] demonstrate binding with its primary targets for deacetylation: lysine residues on the tails of H3 (H3K9) and H4 (H4K16) [3]. In contrast to SIR2-Af1 (the murine homolog of SIRT1), SIRT2 has a lower degree of flexibility in the binding loop, leading to higher substrate specificity despite a wider channel than other sirtuin homologs [45]. Specifically, two residues necessary for deacetylase activity, Asn168 and Asp170, activate the ribose ring to allow for a quicker enzymatic transfer [45] via Ser88 glycosidic attack on the NAD^+^ in SIRT2 thereby allowing the nucleophilic attack on the carbonyl [45]. The catalytic pocket in SIRT2 is deeper but not as wide as SIRT1. Therefore, SIRT2 accommodates longer carbonyl chains rather than bulkier ring structures. He et al. provide proof of SIRT2 cleaving the myristoyl moiety on a lysine chain of H3K9 and TNF-αK19-20 [100]. Further evidence of the larger leaving groups includes SIRT2’s ability for effective de-benzoylation of lysine chain [101].

SIRT2 and SIRT3 are structurally quite similar, though they sit apart from most of the other SIRTs. The amino acid at the -2 position from the protonating residue is not conserved across all sirtuins, but both SIRT2 and SIRT3 share a tyrosine here. Similarly, in the L4 active site loop, a valine at the +7 position from the FGExL loop in both SIRT2 and 3 serves to stabilize the substrate. The −1, −3, and −4 positions from this loop are also conserved between these two enzymes, yet not between any of the other sirtuins.

SIRT2 activity is highly regulated by phosphorylation, largely to modulate the cell cycle checkpoints via decreasing SIRT2 activity. Cyclin Dependent Kinase 1 (CDK1) phosphorylation at S368 or CDK2 at S331 on SIRT2 delays cell cycle progression during the G1/M phase transition [102], whereas CDC14B mediated phosphorylation late in the M/G2 transition may provoke mitotic exit through increasing SIRT2 targeted ubiquitination and degradation. A thorough review by Chen, X, et al. [103] of SIRT2 structure and function with physiologic implications presents insights into the wide-ranging influence of SIRT2. This further highlights the difficulty in the selective targeting of SIRT2 due to its dual role. On the one hand, it demonstrates promise through inhibition for neurodegeneration, while on the other hand, it may rescue some axonal dysfunction, autophagy dysregulation, and neuroinflammation.

SIRT2 is a key regulator of cell cycle progression through the deacetylation of histone core protein 4, lysine 16 (H4K16), and α-tubulin [104]. SIRT2 and HDAC6 work in concert for much of the microtubule regulation and a-tubulin deacetylation; to this end, the first collaborative inhibitors have been developed targeting both SIRT2 and HDAC6 [105,106]. HDAC6 has demonstrated a key role in mood and anxiety via a-tubulin regulation within serotonergic neurons of the dorsal raphe nucleus [107] but the interaction with SIRT2 ties this more closely to cellular energy levels due to SIRT2’s dependence on NAD^+^ [108]. This opens new lines of investigations to target SIRT2 as well as a new pharmacologic point for intervention. Following SIRT2 inhibition, there is a particular subset of perinuclear microtubules, not deacetylated by HDAC6 and, therefore, dependent solely upon SIRT2 for regulation [109].

### 4.2. Psychiatric Implications

#### 4.2.1. Depression

Studies demonstrate that SIRT2 may aid in depressive behaviors as well, such that selective SIRT2 inhibition reduces anhedonia in rodent models of depression via the vesicular glutamate transporter1 pathway (VGLUT1) [110]. This is most notable within the hippocampus. Chronic unpredictable stress produces a depression-like phenotype, which is ameliorated by SIRT2 inhibition within the hippocampus but not the prefrontal cortex or dentate gyrus [111]. Combined, studies surmise that SIRT2 inhibition improves depression-like behaviors via glutamatergic and serotonergic signaling changes [112]. There is evidence to support a proactive role for SIRT2 inhibition in depression across multiple domains and animal models, including a model of depression with reduced olfactory sensation [113]. Unexpectedly, a study of SIRT2 changes after chronic unpredictable stress and without pharmacologic intervention associated symptoms with lower levels of SIRT2 and lower synaptogenesis [114]. Through preclinical studies, there is growing evidence for SIRT2 inhibition promoting CREB and BDNF upregulation, thereby allowing further synaptic density growth [115]. It is thought that SIRT2 may collaborate with SIRT1 in the pro-inflammatory cascade in depression, and inhibition of SIRT2 may aid in ameliorating downstream changes in MDD [14]. These findings underline the need for further study of SIRT2 in anxiety and depression to better formulate therapeutic modalities.

#### 4.2.2. Psychosis and Mania

Sirtuin 2 (SIRT2) has been implicated in schizophrenia, particularly through its role in oligodendrocyte development and myelination. A study by Narayan et al. [116] found that SIRT2 expression was significantly reduced in the prefrontal cortex of schizophrenia patients. Pharmacological inhibition of SIRT2 in mice led to behavioral deficits and myelination abnormalities reminiscent of schizophrenia. These findings suggest that SIRT2 could be involved in the pathogenesis of schizophrenia and that modulating its activity might offer therapeutic benefits.

## 5. SIRT3

### 5.1. Structure and Function

Unlike SIRTs 1 and 2, SIRT3 is localized in the mitochondria and acts primarily as a metabolic protein deacetylase rather than a true histone core protein deacetylase [117]. Examining the surface residues for SIRT3 through in-silico modeling demonstrates that the cap residues are preferred by SIRT3. Investigation of many of the natural client proteins for SIRT3, the side chain and cap residues will provide valuable information for unique substrate synthesis for SIRT3 [118]. According to this model study, SIRT3 prefers the positively charged and aromatic residues on both sides of the modulated lysine, namely, tyrosine, phenylalanine and tryptophan [118]. Many SIRT3 synthetic substrates are successful with a fluorescent compound amino-methylcoumarin (AMC) on one side of the lysine. SIRT3 catalysis relies on substrate cap positioning and binding, allowing the leaving group to be correctly positioned [119]. Di-lysine cap groups also enhance SIRT3 catalytic activity regardless of changes in the leaving group [120]. However, the most selective leaving group for SIRT3 is the crotonyl moiety, as this sirtuin is the only one capable of successfully cleaving a crotonylated lysine moiety [121]. Thus, to construct a SIRT3 selective substrate, it is important to target both the cap as well as the leaving group.

### 5.2. Psychiatric Implications

SIRT3 is directly involved in the regulation of the TCA cycle and fatty acid synthesis through increasing pools of acetyl-CoA molecules by deacetylation and subsequent activation of acetyl-CoA synthetase (ACS) Lys-642 [122] and plays a significant role in the powerhouse of the cell and its functions. SIRT3 and 5 are of interest when identifying potential new therapeutic targets for influencing the cellular energy stores [123]. It is interesting to note that ACS may (has only been reported in yeast thus far) also facilitate the transfer of acetyl groups to the lysine residues of histone core protein tails, activating DNA transcription in like histone acetyl transferases [119,124]. Additionally, SIRT3 influences fatty acid oxidation and breakdown through regulation of the long-chain acyl-CoA dehydrogenase [120].

To date, the metabolic and mitochondrial markers of psychiatric illnesses are not well understood and would benefit from further investigation. Due to the metabolic implications of SIRT3, clinical trials on intermittent fasting to affect aging and oxidative stress, as well as studies of vitamins C and E on SIRT activity, are currently in progress with a specific focus on SIRT3 and interactions with SIRT5 (NCT02132091, NCT02011906).

#### Depression

Additionally, SIRT3 works in conjunction with SIRT1 and is highlighted in studies of resveratrol and its effect on depression preclinically, particularly when associated with mitochondrial dysfunction [25,70]. Antidepressants targeted specifically for peri-menopausal depression, such as kaempferol, produce pro-antioxidant effects and increase the deacetylation of superoxide dismutase 2 (SOD2) [125]. These data demonstrate that Kaempferol’s mechanism of action may be through increasing SIRT3 within the mitochondria. Targeted increases in mitochondrial SIRT3 activity in the hippocampus produced a similar anxiolytic and anti-depressant-like effect [125]. It is thought that SIRT3 may be a major regulator in the activation of the mitochondrial antioxidase to ameliorate depression associated with energy changes, such as that of menopause. Similarly, melatonin also appears to work through a mechanism whereby SIRT3 is activated via mitochondrial free radical oxidation to activate the nuclear factor erythroid 2-related factor 2 (Nrf2) pathways [25,126]. Upregulation of the Nrf2 pathway leads to cellular reduction in free radical oxidation via increasing glutathione and associated reductases. Therefore, SIRT3 appears to work as part of a larger pathway to cooperatively lessen reactive oxygenated species-induced cellular inflammation.

Also, SIRT3 plays a part in the control of the TCA cycle through the deacetylation of the alpha subunit of E1 on pyruvate dehydrogenase (PDH) and ICDH2 K75 and K241 [127,128,129]; while working in conjunction with SIRT4 to regulate the activity of PDH complex. Following unpredictable stress protocols, rodent hippocampal expression of SIRT3 decreased, reactive oxygenated species increased, and the expression of NLRP3 inflammasome as well as interleukin-1β and interleukin-18 increased [130]. Within these rodent models of depression, using chronic unpredictable stress, animals demonstrate altered SIRT3 expression, and symptoms are ameliorated (at least partially) by upregulation of SIRT3 expression and lipid metabolism [131].

## 6. SIRT4

### 6.1. Structure and Function

SIRT4, a lesser-known sirtuin, localizes to the mitochondria and functions primarily as a lipoamidase rather than a deacetylase and retains very little deacetylase activity. Primarily, SIRT4 is responsible for catalyzing the removal of lipoyl and biotinyl moieties from lysine residues on metabolic client proteins [128]. Through delipoylation of pyruvate dehydrogenase E2 subunit, SIRT4 works in conjunction with SIRT3 to regulate PDH activity, suggesting that PDH is regulated by other cofactors outside of E1 phosphorylation, as previously understood [127]. Additionally, SIRT4 aids in cellular metabolism regulation by decreasing reactive oxygen production and increasing ATP production [131,132].

### 6.2. Psychiatric Implications

SIRT4 may play pivotal roles in mitochondrial disease and aging, though more research is currently needed to understand the cellular mechanisms at play. There are few published papers on the role of SIRT4 within psychiatric diseases, though there is a possibility that SIRT4 may play a role in the regulation of mood symptoms through its metabolic effects, as noted above. SIRT4 is thought to be one target of selective serotonin reuptake inhibitor function [133,134]. Furthermore, there is evidence that SIRT4 acts in concert with other mitochondrial sirtuins (SIRT3 and SIRT5), indicating a need to develop pharmacologic agents targeting this enzyme.

## 7. SIRT5

### 7.1. Structure and Function

SIRT5 is the third of the mitochondrial localized sirtuins and regulates proteins involved in oxidation/reduction reactions, fatty acid metabolism, aerobic respiration, and TCA cycle [135]. Like SIRT4, SIRT5 contains very little deacetylase activity and primarily cleaves succinyl and glutaryl moieties from lysine residues [135]. SIRT5 uses a histidine as the deprotonating residue, His158, which is common amongst the sirtuins. However, SIRT5 has a unique arginine in the catalytic pocket [135]. The Tyr102 and Arg105 in the active site allow SIRT5 to cleave succinylated substrates [135]. An interesting study demonstrated the corresponding changes in SIRT5 *Km* and *Kcat* after the addition of various groups at the beta carbon of the succinyl and glutaryl substituents. The conclusion that large *km* changes did not necessarily reflect *kcat* changes demonstrates a fundamental difference in kinetic measurements of enzymatic activity of SIRT5 [136]. Furthermore, the latter study has identified multiple different di-carbonic acyl groups, which can be covalent lysine modifications for carbamoyl phosphate synthetase 1 (CPSI). Thus, it has been found that SIRT5 acts on many di-carbonic acyl chains of 2 to 8 carbons in length [136] as the Arg105 residue in SIRT5 conforms to allow larger acyl residues to interact with the Tyr102. This study also confirms the previous finding that SIRT5 is not an active deacetylase, as the *Kcat* and *Km* for the deacetylase activity is extremely low.

SIRT5 has a further differentiated substrate-docking site due to its two arginine residues in the bottom of the binding pocket. Du et al., [135] demonstrated effectively that these two arginine residues form hydrogen bonding with the carboxylic acid present at the end of the succinyl moiety. Because of this alteration in the active site, the substrate specification for SIRT5 is very straightforward; this is the only enzyme that will easily cleave these terminal carboxylic acid chains. Tan et al. [134] have also observed that SIRT5 is able to act as a deglutarylase on specific proteins with lysine-glutarate residues. Literature reports have suggested that SIRT5 is the only sirtuin with the ability to act as a deglutarylase [134]. The cleavage of succinyl moiety from a lysine residue of client proteins by SIRT5 may play a role in directly regulating the succinyl-coA pool available for TCA cycle. SIRT5 also plays a critical role in mitochondrial and cellular health by controlling reactive oxygen species (ROS) through desuccinylation and activation of superoxide disumutase 1 (SOD1) [28].

### 7.2. Psychiatric Implications

Within MDD, individuals have reduced antioxidant capacity, which is correlated with increased deficits in neurocognition [15]. Manganese superoxide dismutase 2 (SOD2) localizes to the mitochondria and helps to protect the mitochondria from oxidative stress through the catalysis of reactive oxygenated species into more stable molecules. Data demonstrates that individuals with MDD have lower levels of SOD2, and the degree of deficiency correlates with a deficit in cognitive function [15]. SOD2 is highly regulated by SIRT5 through desuccinylation within the mitochondria; this may play a part in the pathophysiology of neuronal metabolic deficits in MDD. Additionally, SIRT5 is upregulated in conjunction with SIRT1 under conditions of calorie restriction [137] and may improve longevity through energy store regulation by desuccinylation of CPSI lysine substrates [135,138]. These changes regulate much of the oxidative phosphorylation chain in the mitochondria, thus affecting the energy of the cell. Downstream, these may correlate with changes to SIRT1 function as well based on NAD^+^/NADH ratio changes.

SIRT5 may also play a role in apoptosis and electron transport chain function through the regulation of other mitochondrial client proteins, such as cytochrome C [129]. Depression behaviors correlate with mitochondrial dysfunction and implicate SIRT5 in this pathophysiology, particularly in the citric acid cycle and oxidative phosphorylation pathways [123]. Reports indicate that SIRT5 may be responsible for the regulation and possible activation of cytochrome C through deacetylation in the inner mitochondrial space (IMS) [129]. Because of these findings, others have investigated the role of SIRT5 in longevity as well, citing energy store fluctuations as a possible implication for early aging [139].

Mitochondrial dysfunction and increased reactive oxygen species with oxidative damage are frequently implicated in schizophrenia pathophysiology [140]. This provides a link to SIRT5. A study of single nucleotide polymorphisms in schizophrenia found no significant interactions between SIRT5 and Acyl-Coenzyme A synthetase long-chain family member 6 (ACSL6); however, a large matching study was proposed and may yield future results for psychiatric illness [141]. While the role of SIRT5 in many diseases has not been fully elucidated, the SIRT5-mediated regulation of many critical client proteins indicates that selective targeting of SIRT5 for the treatment of various pathologies may be beneficial.

## 8. SIRT6

### 8.1. Structure and Function

SIRT6 is more unique in structure than the other sirtuins, though it is less well studied in terms of its function in pathophysiology. While all other sirtuins contain the FGE loop, named for the conserved FGEXL residues, SIRT6 instead contains a loop with WEDSL residues. Along with this, SIRT6 lacks the salt bridge that normally constrains the two loops used in binding NAD^+^ and substrate in the Rossman fold. It has been documented in most sirtuins that the conserved FGExL loop forms the hydrogen bonding interactions with the substrate in the pocket [17]. However, SIRT6 is the only enzyme without these conserved residues; instead, it has a large gap in the protein sequence and corresponding cleft in the quaternary structure, followed by a WEDSL loop. These changes make the selectivity of the cap in SIRT6 much more important. This gives the enzyme more flexibility in its binding mechanism. It is implied by Hu et al. that the cap of the substrate plays a large role in the ability of an enzyme to cleave the lysine substrate. In their study, they tested three versions of TNF-a with a myristoyl-lysine contained within position 20. The cap group and TNF-α version changed the *K_m_* significantly, where the full-length protein was much more effective than truncated cap groups [142]. Further, SIRT6 can catalyze demyristoylation in vitro for TNF-α but cannot effectively cleave myristoylated lysine 9 in histone core protein 3 [142]. However, the same group has also published findings that SIRT2 can also cleave a myristoyl from H3K9 but not from TNF-α [100]. The TNF-α protein contains lysine–lysine moiety; however, the H3 protein contains only one lysine in the peptidic sequence. SIRT6, therefore, requires a Lys–Lys structure for cleavage to occur.

### 8.2. Psychiatric Implications

SIRT6 is recognized for regulating cell cycle and cellular functions through the control of client proteins, such as NF-KB, HIF-1α, and other genes involved in metabolism and aging [143]. It deacetylates H3K9 on the telomeric chain and is involved in the regulation of the aging process through decreasing senescence and double-strand DNA breaks [144]. While SIRT6 does act as a deacetylase, reports have demonstrated that this enzyme may contain demyristoylase activity as well. Though experimentally, the ability of SIRT6 to cleave a myristoyl lysine from H3K9 or TNF-α K19-20, it has not been validated that these activities occur in vivo [145]. Studies indicate a role for SIRT6 in self-regulating the SIRT3 and SIRT4 within the mitochondria, thereby implicating this enzyme as a ‘master regulator’ of mitochondrial energy stores and function within the brain [146]. SIRT6 may be one of the primary regulators of overall mitochondrial DNA replication and expression, thereby altering mitochondrial ROS production and mitochondrial decay. Mitochondrial decay is most prominently noted in telomere shortening, which contributes to aging [143,147].

Pathologically, SIRT6 has been implicated in diseases such as type II diabetes [148] and through this may play a role in the comorbid psychiatric illness with diabetes, such as MDD. SIRT6 has also demonstrated a protective role for the cell during aging [149]. Furthermore, rat data demonstrates a direct link between SIRT6 decline and amphetamine use, thus producing increased hippocampal aging and loss of function [150]. SIRT6 may aid selectively in contextual fear memory and fear learning within the prefrontal cortex, where genetic knockouts experience fear memory learning difficulties without spatial learning issues [151].

## 9. SIRT7

### 9.1. Structure and Function

SIRT7, the most recently discovered sirtuin, is implicated in the cellular transformation from benign to malignant within cancer cells. SIRT7 localized to the nucleus and deacetylates both histones and other non-histone client proteins, though the primary target for SIRT7-mediated deacetylation is H3K18 [152]. SIRT7 plays a vital role in DNA transcription and protein translation through the deacetylation of H3K18 and interactions with non-coding RNAs to regulate rRNA and tRNA synthesis.

### 9.2. Psychiatric Implications

One hypothesis for the change in cellular roles of SIRT7 is that enzyme regulation may come from free fatty acids in the cell, indicating a link between SIRT7 and cellular metabolic status [152,153]. Cocaine use causes increased SIRT7 expression in astrocytes, though this has not been reproduced with other cases of amphetamine use [154]. While SIRT7 remains a promising target for the therapy of cancers, more work is needed to understand the correct pharmacomodulation (i.e., activation or inhibition) that is most beneficial for a patient.

## 10. Summary

Herein, we have summarized the sirtuin mechanism of action and examined the similarities and differences between the enzyme structure and specificity for pharmacologic ligand targeting. We then explored the literature on sirtuins in neuropsychiatric illness across all illnesses. Most of these data are derived from preclinical studies due to the difficulty of studying these enzymes in humans in vivo. SIRT1, 2, 3, and 5 are the most widely implicated in psychiatric illnesses, including roles in substance use, longevity, mood, anxiety, and psychotic illnesses. However, still very little is known about some of the sirtuins, such as SIRT7, and their role in the body and brain. These enzymes play critical roles in both day-to-day cellular function and long-term neuronal disease progression. It remains difficult to singly target individual sirtuins with inhibitors or activators due to the overlap across enzymes in structure and function in areas of the brain and body. Before successful enzyme-selective pharmacotherapies can be developed, more research is needed and will aid in the understanding of the spectrum between norm and disease within the human brain.

## Figures and Tables

**Figure 1 cells-13-01644-f001:**
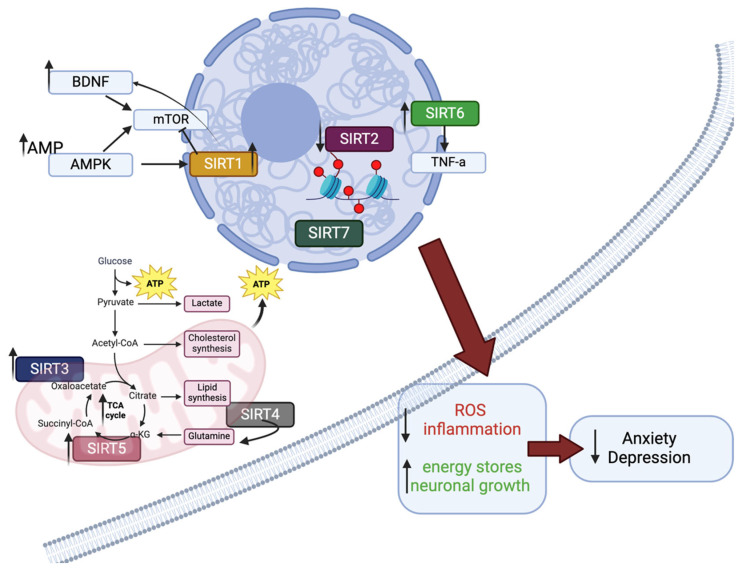
Simplification of the complex role of sirtuins in psychiatric illness, primarily anxiety and depression. This graphic depicts the major roles of each sirtuin within the cell that relate to the cellular energy level and overall mood symptoms. In general, increasing SIRT1, 3, or 5 will increase cellular energy metabolism and decrease mood symptoms, whereas decreasing SIRT2 tends to improve mood through reduced histone deacetylation in the nucleus. This also depicts the localization of SIRTs, where SIRT1,2, 6, and 7 are predominantly nuclear-based, and SIRT3, 4, and 5 are mitochondrial proteins. Abbreviations listed: brain-derived neurotrophic factor (BDNF), adenosine monophosphate (AMP), adenosine monophosphate kinase (AMPK), tumor necrosis factor-alpha (TNF-a), reactive oxygenated species (ROS), tricarboxylic acid cycle (TCA). Created in BioRender. Bonomi, R. (2024) BioRender.com/j83x946.

**Figure 2 cells-13-01644-f002:**
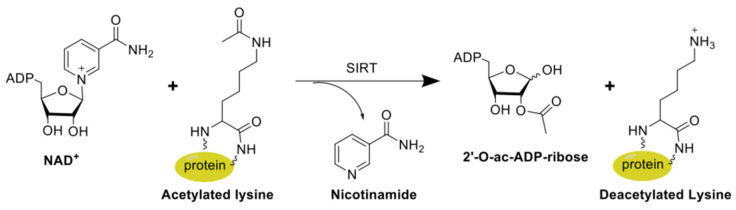
Graphical depiction of the sirtuin mechanism demonstrating the cleavage of nicotinamide adenine dinucleotide (NAD^+^) in conjunction with lysine moiety from the acetylated protein.

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
