# Peer review of "The Structures, Functions, and Roles of Class III HDACs (Sirtuins) in Neuropsychiatric Diseases"

_cells, 2024, doi:10.3390/cells13191644_

Round 1

Reviewer 1 Report

Comments and Suggestions for Authors

The review entitled: “The structures, functions, and roles of HDACs class III (sirtuins) in neuropsychiatric diseases” idea is excellent because it addresses an important topic. However, I realized the review paper was confusing when I read it. Below you can find some observations:

- it is worth mentioning what activates these enzymes, including stress,

- headlines: SIRTn structure should be amended, as they are not just about the structure of these enzymes, but also about their function,

- this review paper is about neuropsychiatric disorders, as the title indicates, and there is a description of conditions that are not neuropsychiatric disorders,

- perhaps it would be worth making subsections in each section on the role of each SIRT in psychiatry for specific disease entities,

- the text should be in the same font,

- were the figures prepared by the author himself or modified?

- Figure 1 - the author should elaborate on all abbreviations and include them in the legend,

- the author uses the same phrases nearby. For example: promise for; page 2, line 58 and 63,

- disproportionate quotation in the text: page 4, line 127; page 6, line 220; page 7, line 282; page 11, line 468; page 12, line 520, 

- excessive or missing spaces: page 4, line 101; page 5, line 163; page 7, line 298; page 8, line 349,

- some sentences are too long: page 6, lines 214-217,

- I would dispense with writing the names of brain structures in italics,

- maybe it is worth explaining the abbreviation in the first quotation in the paragraph: page 11, line 481,

- the word Sirtuins should be written in lower case: page 11, line 508,

- the letter T at the beginning of a sentence should be written without bolding: page 13, line 583.

Reviewer 2 Report

Comments and Suggestions for Authors

Bonomi, Riordan and Gelovani present an interesting review focusing in sirtuins and their roles in psychiatric disorders. For the most part I found the work compelling. My main criticism for the manuscript would be the lack of figures or tables to improve how information is conveyed.

In addition, SIRTs by themselves are a rather divisive subject. In several instances it is difficult to assess if activation or rather inhibition would be more suitable as a therapeutic venue. I ponder if such would be the case in this context.

Comments on the Quality of English Language

English quality is fine, my only suggestion would be edits that improve the flow of the manuscript

Reviewer 3 Report

Comments and Suggestions for Authors

This is a well written manuscript describing possible roles of HDACs class III in the pathogenesis of psychiatric diseases along with their potential therapeutic applications. However, there are some errors/issues that should be corrected before publication.

 -Lines 241-242, please define the direction of change instead of “altered” in the following sentence: Kishi et al. [70] found that SIRT1 expression was altered in the postmortem brains of individuals with bipolar disorder

-Line 252: It appears that several words are missing before the word “Through”.

-Line 278: The word “performed” should be deleted after reference [87].  

-Lines 294-5: The authors stated that “SIRT1 expression, via dihydromyricetin and resveratrol, were protective against deleterious effects of SIRT1 in the liver and the brain”. Both cited references indicate that alcohol decrease SIRT1 and dihydromyricetin or resveratrol increases SIRT1. Therefore, it should be corrected to “against deleterious effects of alcohol in the liver and the brain” (not SIRT1).

-Line 280: Piracetam, although cholinergic, is also known to be a glutamatergic drug. It would be more accurate to describe it as a “cholinergic and glutamatergic agonist” or drug.

-Line 328: It should be “A thorough review by Chen X, …)

-Line 356: Since the results of citation [109] conflict with the results of other studies mentioned before and after this citation, the word "unexpectedly" better suits instead of "Interestingly."

Finally, to provide a rationale for this review, it would be informative and reasonable to mention in the introduction that valproate, extensively used in psychiatric diseases, as well as SCFAs like butyrate, do not affect class III HDACs

Round 2

Reviewer 1 Report

Comments and Suggestions for Authors

Dear Authors,

each sirtuin should be described the same way as the first e.g. SIRT...... Structure and Mechanism of action, SIRT.... Function and........

Author Response

Reviewer 1 comment:

1) each sirtuin should be described the same way as the first e.g. SIRT...... Structure and Mechanism of action, SIRT.... Function and........

Dear Reviewer, 

The formatting and language for headings and subheadings has been improved, such that the wording is more consistent and the headings and subheadings are appropriately differentiated from each other and the surrounding text. Each sirtuin has a section labeled "SIRT1" etc, with subheadings for "Structure and Function" and "Psychiatric Implications" and third tier subheadings with individual disease processes where applicable (i.e. "Depression", "Psychosis and Mania" etc). However, as there is not sufficient data for all sirtuins in each disease process category, some enzymes lack some of these third tier headings. Thank you for your comments, this has improved the readability of the manuscript.